# Estuarine hurricane wind can intensify surge-dominated extreme water level in shallow and converging coastal systems

Mithun Deb[1], James J. Benedict[3], Ning Sun[2], Zhaoqing Yang[1,4], Robert D. Hetland[2], David Judi[2], and Taiping Wang[1]

[1]Marine and Coastal Research Laboratory, Energy and Environment Directorate, Pacific Northwest National Laboratory, Sequim, WA 98382, USA
[2]Energy and Environment Directorate, Pacific Northwest National Laboratory, Richland, WA 99354, USA
[3]Los Alamos National Laboratory, Los Alamos, NM 87544, USA
[4]Department of Civil and Environmental Engineering, University of Washington, Seattle, WA 98195, USA

**Correspondence:** Mithun Deb (mithun.deb@pnnl.gov)

**Abstract.** Based on the projected increase in hurricane landfall frequency on the middle to lower U.S. East Coast, we examined the crucial role of the estuarine wind field in exacerbating coastal flooding. A regionally refined atmospheric and two high-resolution hydrology and ocean models are integrated to provide plausible and physically-consistent ensembles of hurricane events and the associated flooding inside the Delaware Bay and River, a U.S. mid-Atlantic estuary. Model results show that the hurricane propagation direction, estuarine geometry, remote surge from the open ocean, and direct nearshore upwind stress could magnify the flood magnitude. More specifically, inland-bound tracks that make landfall before reaching the mid-Atlantic coast produce a more significant surge within Delaware Bay than the shore-parallel tracks, where the estuarine wind direction plays the primary role in surge amplification. Ultimately, this study emphasized the need for integrated models to capture the nonlinear dynamics and interactions in flood hazard modeling.

## Plain Language Summary

This study examines how estuarine wind field can exacerbate hurricane-driven coastal and riverine flooding. We coupled earth system, hydrology, and hydrodynamic models to generate plausible and physically consistent ensembles of hurricane events and their associated water levels from the open coast to tidal rivers of Delaware Bay and River. Our results show that the hurricane landfall locations and the estuarine wind can significantly amplify the extreme surge in a shallow and converging system, especially when the wind direction aligns with the surge propagation direction. Other essential factors, such as the amplitude of the remote surge from the open ocean and the magnitude and timing of peak river discharge, can further amplify the extreme water level. Ultimately, this work demonstrated the need for integrated modeling to capture the wind, tide, and surge interactions during a hurricane landfall and why such a framework is critical for improving coastal hazard projections in a warmer climate.

## 1 Introduction

Due to their low-lying topography and high population densities, coastal cities and estuaries are particularly susceptible to storm surges, which result in significant economic and social impacts like damage to infrastructure, loss of property and livelihoods, and loss of life (Dietrich et al., 2010; Bilskie et al., 2016; Valle-Levinson et al., 2020). Recent climate studies have indicated that, under different greenhouse gas emission rates and global warming scenarios, there is a potential for more frequent hurricane landfalls on the U.S. Atlantic coast at the end of the 21st century (Knutson et al., 2022; Balaguru et al., 2023). Some other works have shown a potential increase in hurricane's maximum intensity (Emanuel, 2005, 2021), an increase in the 10 m wind speed (Roberts et al., 2020), and a decrease in translation speed (Emanuel, 2021; Garner et al., 2021) in the future. These projected trends in hurricane characteristics can amplify the risk faced by coastal cities and estuaries to storm surges and coastal flooding in the future climate. More recently, using Coupled Model Intercomparison Project Version 5 (CMIP5) global climate model datasets, Weaver and Garner (2023) examined the hurricane landfall patterns in the U.S. East Coast for a warmer climate. They showed a positive trend in hurricane genesis points moving northward and making more landfalls along the mid-Atlantic region as they traveled through the U.S. Northeast (similar to Hurricane Irene 2011). This is a matter of concern for the U.S. mid-Atlantic region that covers two of the largest estuaries in the U.S.: the Delaware and Chesapeake Bays. These estuarine regions house $\sim 27$ million inhabitants, a large density of metropolitan areas, natural ecosystems (e.g., salt marshes and freshwater wetlands), transportation networks, and industrial ports (Callahan and Leathers, 2021). The hurricane landfall trend in this area certainly raises interest in the associated storm surge hazard that might help develop flood risk assessment tools for federal, state, and local agencies.

To estimate the hurricane-induced flood risk, significant effort has been invested in storm surge prediction using high-resolution numerical models that resolve fundamental underlying physics (Weisberg and Zheng, 2006; Wang et al., 2008; Hu et al., 2009; Dietrich et al., 2011). However, a significant impediment to accurate storm surge predictions continues to be the uncertainty in predicting hurricane properties that drive storm surge (Cyriac et al., 2018; Cangialosi et al., 2020). More specifically, an inaccurate representation of any of these hurricane characteristics: intensity, size, translation speed, and the angle of landfall with the coast can introduce large biases in predicting the surge and coastal flooding (Suh and Lee, 2018). In a recent study, Hsu et al. (2023) examined the role of these variables for three different hurricanes that propagated through the South Atlantic Bight [Matthew (2016), Dorian (2019), and Isaias (2020)] and showed how they affect the peak storm surge and wave runup in the South Atlantic coastline. For the same region, Parker et al. (2023) demonstrated that various combinations of tide, non-tidal residual, and wave setup and their spatially varying interaction can control the total water level at different U.S. Southeast Atlantic coastline regions. In addition, storm surge also strongly depends on the local geometry of the basin and bathymetric features (e.g., the angle of a coastline), such that an adequate representation of the system's geometry is essential for properly estimating the hydrodynamic response (Weisberg and Zheng, 2006; Resio and Westerink, 2008; Suh and Lee, 2018). Geometry is especially important in estuarine regions, where the storm surge and storm tide (sum of surge and the astronomical tide) propagation and flood generation can vary spatially based on the size and bathymetry of the bay, cross-sectional area, the shape of the system (e.g., funnel-shaped estuary, rectangular shaped tidal lagoons, etc.), and river discharge

from the hurricane-induced precipitation (Mori et al., 2014; Familkhalili and Talke, 2016). In general, in hyper-tidal estuaries (exhibit large tidal range) or convergent estuaries (channel area convergence dominates bottom friction), the variability of tide-surge interaction can amplify the extreme water level (Lyddon et al., 2018). In addition, the magnitude and duration of the estuarine winds during hurricane landfall can exacerbate the spatial extent and amplitude of surge-driven flooding (Shen et al., 2006a; Weisberg and Zheng, 2008).

In shallow estuaries, the local wind field can create a surface slope between the bay mouth and the upstream end (Weisberg and Zheng, 2006, 2008). This evolution becomes much more complex for a convergent system where the non-linear interaction between the tide, surge, and local geometry influences the flow acceleration, ultimately amplifying or damping the surge-induced flooding (Wong and Moses-Hall, 1998). When the remote surge generated by the wind field in the open ocean reaches the coastal environment, it interacts with the estuarine wind and generates a complex set-up or set-down within the estuary (Shen et al., 2006b; Defne et al., 2019). There have been many studies on the storm surge hindcast and long-term flood hazard projections for different coastal or estuarine regions [e.g., Villarini et al. (2014); Wahl et al. (2015); Marsooli et al. (2019); Lin et al. (2019); Bates et al. (2021); Gori et al. (2022)]; however, the effect of estuarine winds on surge amplification during hurricane landfall is not often separately analyzed. For several U.S. coastal regions, Lin et al. (2010) and Marsooli and Lin (2018) used higher-resolution hydrodynamic models to assess the sensitivity of coastal flooding to different storm characteristics, but they also focused primarily on the overall surge evolution during the entire hurricane period and not on the local amplification by the nearshore wind field during landfall.

Similarly, several other studies used the simplified storm surge model: Sea, Lake and Overland Surges from Hurricanes [SLOSH; Jelesnianski (1992)] to assess basin-scale flooding from individual extreme events (Powell and Houston, 1996; Houston et al., 1999) or the ensemble of probabilistic hurricane tracks (Taylor and Glahn, 2008). While these studies extensively discussed the role of landfall location, wind direction, and coastal geometry, the role of estuarine wind in the local amplification of storm surge remained unexplored. Also, it should be noted that in a system with an irregular bottom and rapidly converging width, it is challenging to assess the role of estuarine wind using simplified models like SLOSH due to the absence of advective terms in the momentum equations, exclusion of the river flow and the parametric representation of the hurricane wind field (Glahn et al., 2009). To properly examine the role of estuarine wind and the non-linear interaction in surge amplification in a complex system, an integrated modeling framework with higher-order physics is required that resolves the atmospheric, hydrologic, and hydrodynamic processes during hurricane landfall.

This study aimed to answer, depending on the estuarine wind field (i.e., landfall location and wind direction), how different the hydrodynamic responses during flooding when compared between an inland hurricane (making landfall before reaching Delaware Bay) and a shore-parallel track (making landfall after passing Delaware Bay). This is an important question because of the potential changes in landfall trends in a warmer climate in the U.S. Mid-Atlantic region. We focused on two tasks: 1) integrating an earth system model [E3SM, Golaz et al. (2022)] with high-resolution models that couple the hydrology [DHSVM, Wigmosta et al. (1994)] and hydrodynamic [FVCOM, Chen et al. (2003)] models to provide a high-fidelity flood estimate, and 2) evaluating the role of the estuarine wind field in amplifying storm surge during hurricane landfall. We chose Delaware Bay and River (DBR), a shallow and convergent estuary in the U.S. Mid-Atlantic, historically highly vulnerable

to storm-induced flooding. We also selected Hurricane Irene (2011) as a focal event for the study, which caused one of the most severe estuary-wide flood hazards in DBR. Using E3SM, we first perturbed Hurricane Irene (2011) to get an ensemble of Irene-like tracks with different characteristics. Then using the coupled DHSVM-FVCOM, we predicted the total water level from different tracks and the flood distribution along the entire DBR. Incorporating these three models provided a high-fidelity representation of coupled atmospheric, fluvial, and coastal processes, which allows for improved flood estimates for Hurricane Irene-like events. Finally, we separately looked at the role of the estuarine wind field in amplifying the surge as the tide propagates upstream.

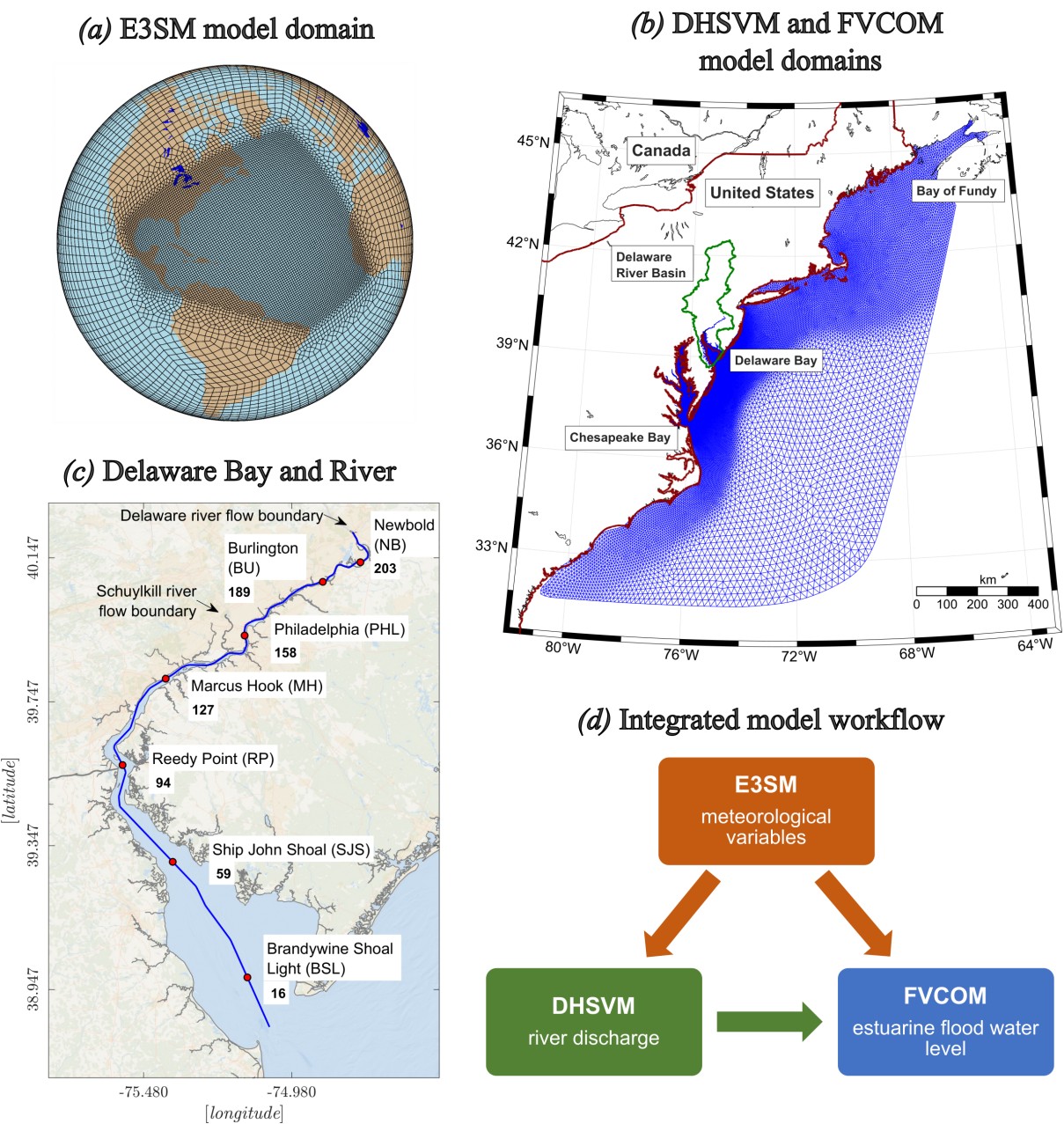

**Figure 1.** (a) E3SM grid resolution: 100-km globally uniform resolution and a 25-km resolution over the North Atlantic; (b) Regional scale model domains, where blue triangles show the FVCOM grid cells and green polygon represent the DHSVM model coverage; (c) Focus area of the study, Delaware Bay and River (DBR). Red circles show tide gauge locations used for FVCOM model validation with their distance from the Bay mouth (in kilometers); (d) Schematic showing the coupling strategy between the different models used in this study.

## 2 Model setup and integration

The U.S. DOE Energy Exascale Earth System Model version 2 [hereafter, "E3SM"; Golaz et al. (2022)] is used to simulate an ensemble of Hurricane Irene-like tracks. E3SM integrations are conducted using prognostic atmosphere, land, and river model components, while oceanic sea-surface temperatures (SSTs) and sea ice cover are prescribed based on observations (Huang et al., 2021). The atmosphere model version used in this study has 71 levels and uses a regionally refined horizontal grid mesh of ~25 km over the North Atlantic basin and eastern North America with a coarser horizontal resolution (~100 km) outside this region [Fig. 1a and as described in and used by Zarzycki et al. (2017)]. The atmosphere model is initialized globally based on ECMWF Reanalysis version 5 [ERA5; Hersbach et al. (2020)], and all simulations span 30 days to cover the atmospheric spin-up and evolution of Hurricane Irene as well as the meteorological conditions in the wake of the event. Additional details of the E3SM setup and validation metrics are found in Appendix A.

We conduct an E3SM ensemble to acknowledge (a) our incomplete understanding of the model physics and (b) the inherent uncertainty of model initial conditions. To address (a), we perturb model physics parameters to which tropical cyclones are most sensitive [He and Posselt (2015); see Appendix A for the list of perturbed parameters and additional information on the model ensemble setup and analyses]. This is accomplished by first defining acceptable numerical ranges for each parameter of interest [based on He and Posselt (2015)] and then randomly drawing 50 values within those ranges to use within the model integration. In this way, 50 unique parameter "sets" are created. To address (b), E3SM is initialized at two times separated by 12 hours [August 26, 2011 00Z (hereafter "E1") and August 25, 2011 12Z (hereafter "E2")]. Both the ERA5-based meteorological patterns and the diagnosed storm center at these two initialization times are slightly different from each other and ultimately result in a more diverse spread of simulated Irene tracks. E1 was identified as the optimal initialization time that produced reasonable Irene tracks at the earliest forecast lead time, while E2 was also retained to understand the sensitivities of the predicted tracks to large-scale meteorological patterns at initialization time. In total, two 50-member ensembles (E1 and E2) are generated (Figure 2a).

Subsequently, we employ the Distributed Hydrology Soil Vegetation Model (DHSVM) to estimate the fluvial flooding in the Delaware River Basin. DHSVM used meteorological variables from the E3SM model as climate forcing, including precipitation, air temperature, downward shortwave and longwave radiation, wind speed, and relative humidity. DHSVM is a process-based, spatially distributed hydrological model that operates at the grid cell level. It simulates key overland and subsurface hydrological processes by solving the full energy and water balance equations. The model physics and formulations have been extensively described in the existing literature [e.g., Wigmosta et al. (1994); Sun et al. (2015); Perkins et al. (2019); Sun et al. (2024)]. For this study, we used a 90-m resolution for DHSVM in the Delaware River Basin and ran the model at a 3-hourly timestep.

Finally, to predict the extreme water surface elevation (WSE) with the atmospheric and riverine forcings, we use the unstructured grid Finite Volume Community Ocean Model FVCOM (Chen et al., 2003), which has been extensively used for storm surge modeling in many estuaries worldwide [e.g., Weisberg and Zheng (2006); Rego and Li (2009)]. We chose the 3D barotropic and hydrostatic version, which resolves simplified Reynolds-averaged Navier–Stokes equations (with Boussinesq

approximations). The model domain extends 700 km offshore from the mid-Atlantic coast to adequately capture the air-sea interaction and 215 km from the Delaware Bay mouth to the river flow boundary (Figure 1b,c). The horizontal grid resolution is assigned to 25 km along the open ocean boundary to 20 m in the river to seamlessly simulate the interaction between large- and small-scale flood processes. Subsequently, we enforced the lateral and sea surface boundary conditions: 1) tidal forcing at the open ocean boundary [WSE from the TPXO8.0 global ocean tide model (Egbert and Erofeeva, 2002)], 2) river flow from DHSVM, and 3) 10-m wind speed and mean sea-level pressure from E3SM. More details about the topography and bathymetry data sets used for grid development, activated model physics, and assigned numerical variables/coefficients are given in Deb et al. (2023).

## 3 Hurricane-induced estuarine flooding

At the onset, DHSVM simulated daily flows (averaged from the 3-hourly flows) were validated against USGS daily flow observations at six gage locations on the main stem of the Delaware River for Hurricane Irene (2011) [Appendix B]. After observing a satisfactory model performance (Nash-Sutcliffe efficiency and the Kling–Gupta efficiency), we simulated the E3SM ensembles that produced a different range of peak river discharge at the Delaware River boundary, shown in Figure 2b. The range and the ensemble mean of the river discharge from the two sets show a clear distinction, where the inland-oriented tracks (E2) produced a smaller magnitude and an earlier peak ($\sim 12\, hours$) compared to the shore-parallel tracks (E1). In this work, as we focused on the response of estuarine flooding to different hurricane characteristics, we directly used the DHSVM data as river forcing for FVCOM without discussing the underlying physical processes responsible for the variation in magnitude and phase of the fluvial discharge.

Before running FVCOM with meteorological conditions from E3SM and river forcing from DHSVM, we also validated the hydrodynamic model by comparing predicted WSE with the observed data for two hurricanes, Hurricane Irene (2011) and Hurricane Sandy (2012), that both had devastating impact on DBR. For both cases, we found that the model successfully predicts the amplitude and phase of the storm surge and shows an excellent agreement between the model and observed data at various tide gauge locations throughout the system [Appendix C]. Thus, the model was deemed fit for purpose for our subsequent studies using ensemble forcing.

The ensemble simulations, forced with two 50-member ensembles of Irene-like cases (E1 and E2), show significant variability ($\mathcal{O}(1\, m)$ standard deviation in WSE) in the range of along-channel peak WSE over the ensemble forcing (Figure 2c). E2, with more inland tracks, generates a much higher range of surge than E1 for a significant portion of DBR; E2 also has a significantly higher ensemble mean WSE, especially in the mid-bay approximately 150 km from the mouth, where the E2-forced ensemble mean WSE is over 2 m higher ($\sim 100\%$ increase) than the E1-forced case. E1, which produced reasonable Irene tracks (Figure A1), also shows a fair range of WSE and an ensemble mean compared to the observed along-channel peak WSE for Hurricane Irene (2011). Here, observed peak WSE means the FVCOM model WSE generated using reanalysis forcing and validated using field datasets. At the upstream end, near NB, the peak WSE range deviates from the observed due to the influence of river discharge and the biases that propagated from the E3SM precipitation field (details provided in Appendix A).

Interestingly, close to the bay mouth and near the upstream river boundary, the distribution of E1- and E2-forced WSE are similar, despite strong differences in the mid-bay. The similarity at the bay mouth indicates that the offshore hurricane wind field (outside the bay) generated a narrow range of the remote surge, which propagated upstream from the bay entrance. Then, there is a local generation of storm surge inside the bay (between BSL and BU), significantly higher for E2, likely due to a combination of the hurricane's estuarine wind field and convergence of the estuarine width. The river discharge plays a secondary role; we can see that for E1, which produces a much higher flux magnitude, the impact is only significant for the region between BU and NB (the narrowest portion near the DHSVM flow boundary). In general, for all the tracks from E1 and E2, the peak river discharge at the Delaware River boundary lagged the storm surge at the bay entrance by two days, making the impact of river discharge negligible during the landfall period on Aug 28, 2011 12Z. The higher surface slope near the river boundary for E1, shown in Figure 2c, was generated primarily from the peak river discharge that occurred around Aug 30, 2011 12Z (Figure 2b). While a similar along-channel gradient can also generate for E2 around Aug 30, 2011 12Z, the significantly higher surge during the landfall (Aug 28, 2011 12Z) elevated the peak WSE much more than the following river discharge-driven condition. A more detailed explanation of this WSE variation is given in the following section.

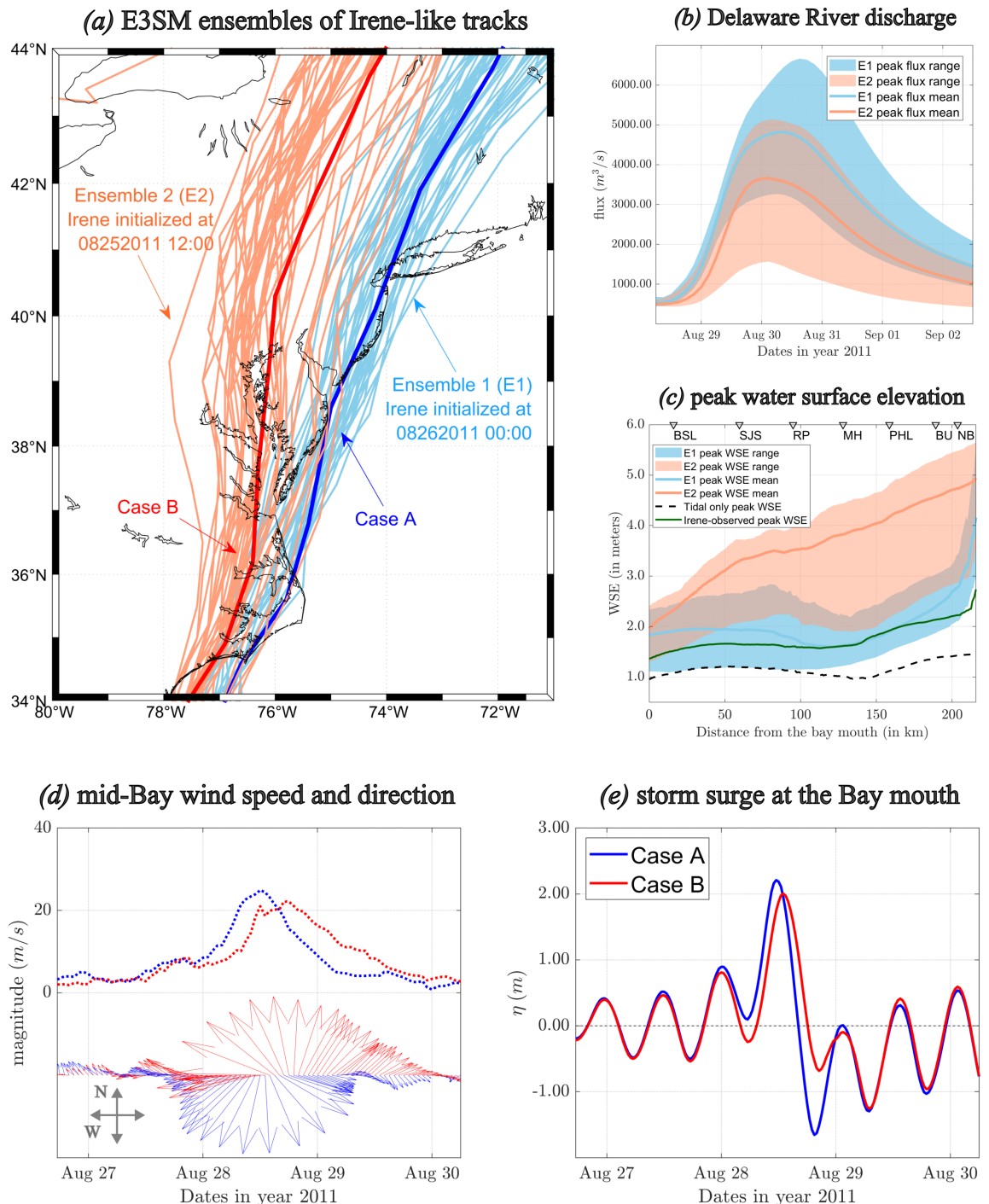

**Figure 2.** (a) Ensembles of hurricane Irene-like tracks generated using E3SM; (b) Delaware River discharge for the tracks in (a) by DHSVM; (c) Predicted peak water surface in DBR using FVCOM. The observed peak water surface elevation during Hurricane Irene is shown using a green solid line; (d) Estuarine wind speed and direction (length of the vectors are scaled using wind speed) for Cases A and B; (e) Storm surge elevation for the same events.

## 4  The role of local estuarine wind in surge amplification

In this section, we examined the impact of estuarine wind directions in amplifying storm surges along the converging DBR system. We selected two tracks from the ensembles – Cases A from E1, and B from E2 (Figure 2a) – that produced a nearly similar surge at the Delaware Bay mouth (the 'remote surge' hereafter), shown in Figure 2e. Cases A and B also have comparable 10-m elevation wind magnitudes inside the bay (Figure 2d), though the strongest winds are in opposite directions; Case A with a primarily northerly wind, and Case B a southerly wind. We provided a more thorough description of different process comparisons that led to these event selections in the supplemental material titled "Hurricane case selection".

Figures 3a,b shows a 2D representation of hurricane wind field for the two events when the estuarine wind speed peaked ($\sim 20\,m/s$) inside DBR. To isolate the role of estuarine wind, we defined a region (the black polygon in Figures 3a,b) that covers the hydrodynamic model domain from the bay mouth to the upstream model boundary. We used this bounding polygon to select the E3SM grid cells that fully cover the FVCOM model domain. FVCOM uses a bilinear interpolation method to assign wind velocity at the unstructured grid cells from the meteorological dataset. To represent a scenario with nominal estuarine wind during hurricane landfall, we multiplied the E3SM wind velocity vectors (in m/s) within the polygon with 0.1 to uniformly dampen the wind magnitude in the selected cells, regardless of the instantaneous location of the hurricane. For the two cases, A and B, this artificial dampening reduced the peak wind speed magnitude to $\sim 2.0\,m/s$, making the impact of the hurricane wind field negligible. In addition, the polygon and E3SM grid cells extensively covered the FVCOM model domain, going beyond the FVCOM boundary, to better interpolate the wind forcing. Figures 3c and 3d show the instantaneous WSE along the main channel for the two tracks based on simulations with and without the effect of estuarine wind fields within the defined region.

It is evident that the local estuarine wind stress plays a significant role in both cases in altering WSE as the storm surge propagates upstream. However, the estuarine wind plays a more prominent role in Case B, where the southerly wind is directed up-channel, and pushes more water northward into the converging portion of the Bay, resulting in a rapid increase in the water level from BSL to NB (Figure 2c). In contrast, Case A shows a markedly different response within the system, as the northerly wind causes a set-down at the upstream portion of DBR and a set-up in the mid-bay resulting from the interaction of the local and remote surge, which produced higher flooding ($\sim 1.0\,m$) between SJS and PHL (Figure 3c).

Previous studies have demonstrated how the local estuarine wind field created surface slope during hurricane landfalls at Tampa Bay, FL (Weisberg and Zheng, 2006), and Chesapeake Bay, VA (Shen et al., 2006b), respectively. They have shown that, in shallow estuaries, the cross-shore component of wind stress creates a large downwind surface slope and surge. To explain this process mechanistically, Wong and Trowbridge (1990) and Shen et al. (2006a) provided linear solutions for WSE along a hypothetical rectangular estuary by dividing the total water level into two parts: remote surge and set-up/set-down from local wind forcing. While the linear and simplified momentum equations (using a constant bottom and width of the estuary) can provide a fundamental understanding of the spatially-varying surge amplitude, the evolution of flooding becomes much more complex for a convergent system. In such cases, the non-linear interaction between the tide, surge, and local geometry may increase or decrease along-channel WSE (Weisberg and Zheng, 2008; Xiao et al., 2021). In a complex system

with irregular bottom and rapidly converging estuary width like DBR, the simplified equations cannot reasonably predict the
surface variation shown in Figures 3c,d, and a high-resolution 3D numerical modeling framework is necessary to address the
non-linear interactions and the associated surge generation.

The evolution of peak WSE for the entire period of the hurricanes was examined by designing a few scenarios to identify the
primary drivers of estuarine storm surge intensity. In the hydrodynamic model run, surface wind stresses (estuarine, remote, and
both) were included along with the tidal forcing in a sequence to explain the incremental flood amplification, shown in Figures
4a,b. As described earlier, we multiplied the E3SM wind velocity vectors (in m/s) by 0.1 outside/within the polygon (shown in
Figures 3a,b) to make estuarine/remote wind stress-only cases, respectively. For Case A (with a northerly wind from E1), the
estuarine wind aligned against the incident tidal wave (without remote surge) and produced a steep slope between MH and SJS
due to the complex tide-wind interaction and channel convergence. The downwind water surface pressure gradient in the estuary
caused additional resistance to the propagating flood tide. Combined with the convergence effect, this non-linear interaction
amplified the bay water level (Figure 4a). When the full wind field is included, we can see a similar along-channel peak WSE
gradient where the difference in peak water level, varying from 0.73 to 1.05 m, came from the remote surge propagation through
Delaware Bay. Figure 4a also shows that the remote surge, as it travels through the system, follows a similar trend of the peak
tidal WSE where some amplitude damping occurs close to RP due to higher flooding of the surrounding low-lying wetlands.
The combined effect of the remote surge and local set-up from the estuarine wind field attributed to a locally amplified flooding
on the bay-river interaction zone (between SJS and MH), where the peak water surface increased by more than $1.0\,m\,(\sim 60\%)$
at certain locations. A spatial map of the peak WSE difference between surge from the full wind field and the remote field
only ($\Delta \eta$ in meters) is given in Figure 4c to demonstrate the extent of the flooding in the same zone. Finally, when the river
discharge from the hurricane precipitation field is added to the full wind simulation, it seems to generate a steep WSE gradient
and compound flooding on the upstream part of the Delaware River (Figure 4a), where the river width is significantly narrower
than the estuary and bay mouth.

Case B, with a southerly wind, shows much higher flooding (almost doubled near the head) for the full simulation than Case
A (Figure 4b). As the estuarine wind and remote surge propagated in the same direction, the direct upwind push of the water
from the large bay surface raised the peak WSE from BSL to the upstream model boundary. While we can see a change in
the water surface slope near SJS from the higher overland flooding, the combined WSE from wind-generated local surge and
remote surge demonstrate a linear increase along the channel. Between BSL and SJS, the estuarine wind and remote surge
interacted similarly to the previous observations of Wong and Trowbridge (1990) and Shen et al. (2006a). However, from SJS
to the remaining narrow reach, the channel cross-section area convergence amplified the peak WSE as the wave propagated
upstream. Compared to the remote surge-only case, the combined estuarine wind and remote surge case produced a 2 m
higher water surface elevation ($\sim 100\%$ increase). The inclusion of river discharge did not make any notable difference in the
upstream regions (as seen in Case A) due to the larger surge-driven channel water volume and surface area. Figure 4d shows
the peak WSE difference ($\Delta \eta$) for Case B, where we see a dramatic increase in along-channel flooding from the estuarine
wind field. This case illustrates that the local estuarine wind stress can significantly amplify WSE from the bay entrance to

upstream regions in converging estuaries (nearly $2.5\,m$ or $150\%$ for Case B), even when the offshore surge near the entrance is

not catastrophic.

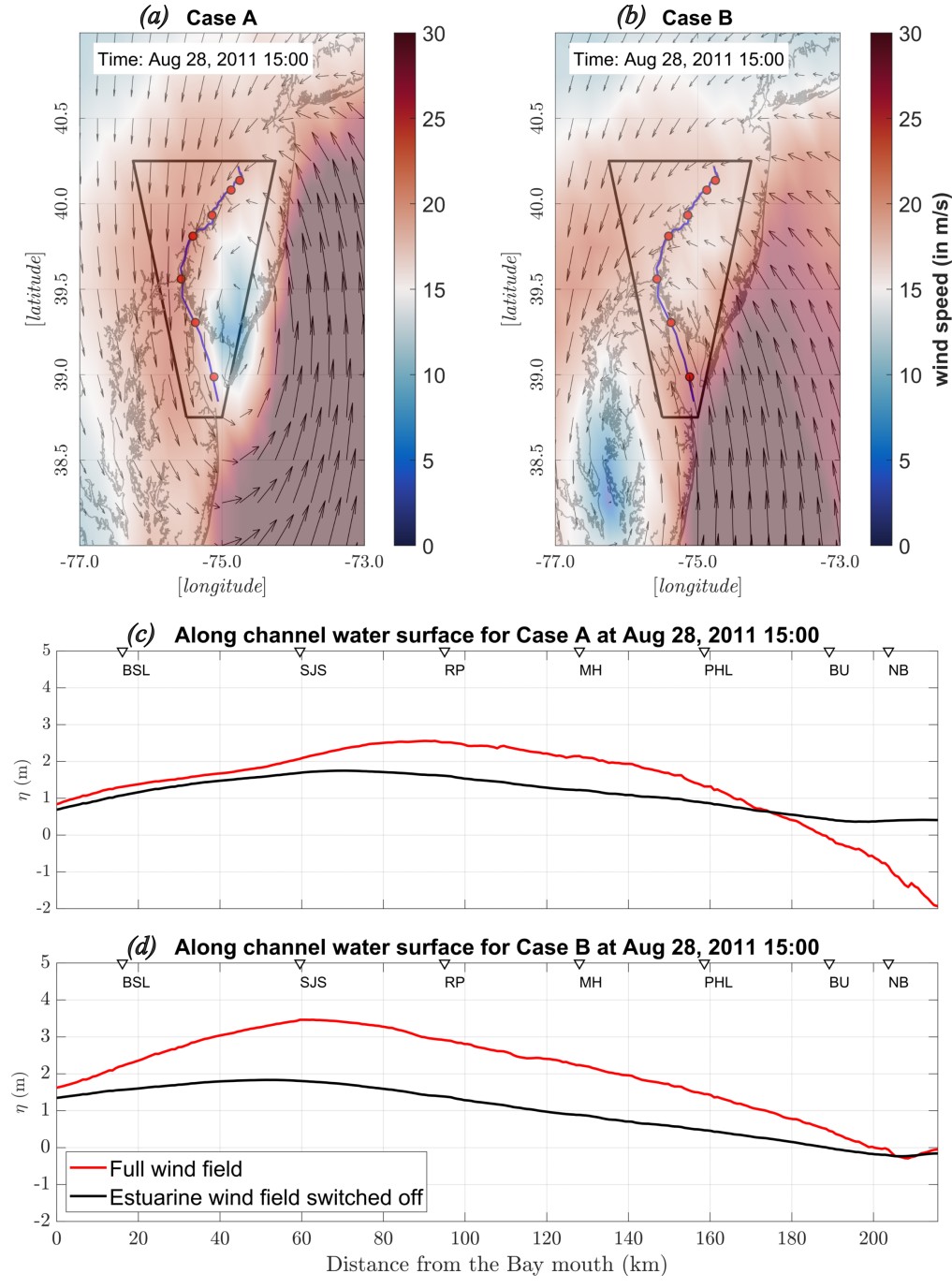

**Figure 3.** (a,b) Instantaneous 10-m elevation wind field during the landfall for hurricane Cases A and B. Polygon shape drawn on top of DBR to represent the region used to dampen the wind stress; (c,d) Comparison of the instantaneous along-channel water surface with and without the estuarine wind for both cases.

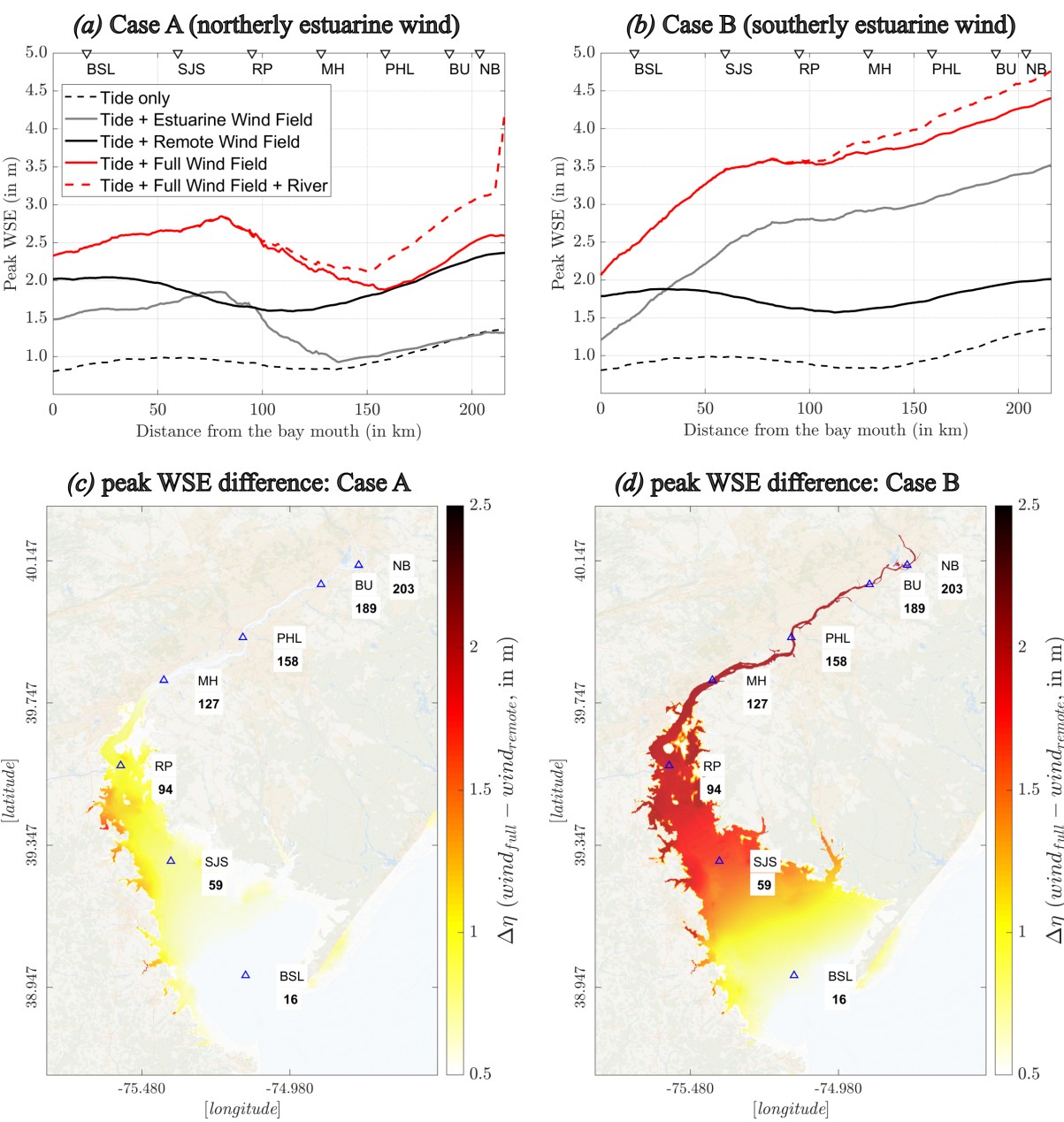

**Figure 4.** (a,b) Changes to along-channel peak water surface elevation for different force combinations for Cases A and B; (c,d) The difference in peak water surface with and without estuarine wind for the same cases.

## 5  Discussion and conclusions

This study examined the role of different hurricane landfall locations and the associated estuarine wind field on the local amplification of storm surges in a converging coastal system. A key finding was that small differences in hurricane tracks could cause drastic differences in the bay surface wind stress (time scale is on the order of hours) and the along-channel storm surge response due to local geometry. Previous works related to the sensitivity of storm surge and coastal flooding to hurricane landfall locations, wind field (speed and direction), and geometry [e.g., Powell and Houston (1996); Houston et al. (1999); Shen et al. (2006b); Weisberg and Zheng (2008); Marsooli and Lin (2018)] have not separately examined the role of this shorter period (translation period through the estuary) and estuarine-scale landfalling wind using physics-based integrated modeling frameworks.

Thus, there is a clear need to better understand the complex, nonlinear response of storm surge to storm track, as simple metrics such as distance to storm track could be misleading and fail to capture the intricacies of the response. Further work is also needed to examine the role of hurricane intensity, the radius of maximum wind, translation speed, and the interaction between tide, non-tidal residual, and waves, separately, all of which could similarly influence the coastal flood level (Suh and Lee, 2018; Parker et al., 2023; Hsu et al., 2023). These essential hurricane characteristics and oceanic processes can affect both the remote and local surge generation, requiring a future study focusing on their overall impact on the same area. Also, this work emphasized the utility of a model hierarchy, which is crucial for accurately representing important localized factors such as the estuarine wind field. This approach is essential for improving future coastal hazard projections on the U.S. Atlantic coast for a warmer climate.

The integrated modeling framework combining an earth system model (E3SM), hydrology model (DHSVM), and hydrodynamic model (FVCOM) helped generate plausible and physically-consistent two ensembles of hurricane Irene-like events, associated river discharge, and the coastal flood water level, respectively. Peak water surface elevation inside Delaware Bay and River, a shallow and convergent system, showed a pivotal relationship with the estuarine wind directions. Even though tracks in Ensemble 2 (inland hurricanes) made landfall much earlier before reaching the bay compared to Ensemble 1 (shore-parallel hurricanes), and both had a similar storm surge magnitude at the bay mouth (propagated from the open ocean), the inland-oriented tracks produced greater flooding inside the bay and river despite much smaller river discharge. Upon examining the flood generation mechanisms step-by-step, we observed that the estuarine hurricane wind could significantly amplify flooding in shallow and converging estuaries when it follows the surge propagation direction. Case B, which has shown this effect, produced nearly $2.5\,m$ or $150\%$ increase in the peak WSE from the bay entrance to the upstream river portion compared to the case with remote surge only. The water surface gradient demonstrated a spatially-varying effect of the non-linear interaction between tide, surge, wind stress, and the estuary geometry, where the channel convergence rapidly raised the peak WSE as the flood wave propagated upstream. When the surge and estuarine wind direction opposed each other (as seen for Ensemble 1), the complex interaction produced a set-down in the river, a set-up in the bay, and a higher surge in the middle of the estuary. As shown for Case A, while the flooding is limited to the mid-portion of the estuary, the peak WSE increased again by more than $60\%$ compared to the case without bay surface wind stress.

Ultimately, the analysis showed that even if a hurricane makes landfall before reaching the mid-Atlantic region and does not bring a record extreme water level near the bay mouth from offshore, the direct upwind forcing over the estuarine surface area itself can produce a record level of flooding if it aligns with the surge direction. In converging estuarine systems worldwide [e.g., the Delaware Bay and River (USA), Humber Estuary (UK), Hooghly Estuary (India), the Meghna River Estuary (Bangladesh), and the Pearl River Estuary (China)] that are highly vulnerable to hurricane-induced flooding, physically consis-

tent and integrated modeling frameworks are critical to correctly resolve this nonlinear tide-wind-surge dynamics and improve the coastal hazard projections for a future climate. In other coastal systems, such as sheltered tidal lagoons or river deltas, properly resolving the estuarine local wind using an integrated framework is essential as well; however, the interacting effect of geometry and tide-wind-surge dynamics in flood amplification will be less significant than the converging ones.

*Code and data availability.* ECMWF Reanalysis version 5 data were obtained from https://doi.org/10.24381/cds.bd0915c6. NOAA OI-SST

version 2 data were obtained from: https://www.psl.noaa.gov/data/gridded/data.noaa.oisst.v2.highres.html. The E3SM version 2 code base is accessible via a Github repository at https://github.com/E3SM-Project/E3SM. The 3D ocean model FVCOM code is available from the MEDM Lab (https://github.com/FVCOM-GitHub). The Distributed Hydrology Soil Vegetation Model (DHSVM) code is available at https://www.pnnl.gov/projects/distributed-hydrology-soil-vegetation-model. Hydrodynamic model tidal boundary conditions are assigned using OSU TPXO Tide Models (https://www.tpxo.net/home), and model validations are performed using tidal water level from NOAA

tides and currents (https://tidesandcurrents.noaa.gov/). E3SM ensemble setup and management was facilitated using the Betacast software package (https://github.com/zarzycki/betacast). All E3SM-simulated tracks and storm characteristics are computed using the TempestExtremes software package (https://github.com/ClimateGlobalChange/tempestextremes), and model errors in storm location and intensity compared to the IBTrACS observation-based data set are obtained using version 10.0.0 of the Model Evaluation Tools Tropical Cyclone (MET-TC) diagnostic package (available at https://github.com/dtcenter/MET). Model data sets used for further analysis are provided here at

https://doi.org/10.5281/zenodo.7988098.

## Appendix A: E3SM setup and validation

The U.S. DOE Energy Exascale Earth System Model version 2 [hereafter, "E3SM"; Golaz et al. (2022)] land model shares the same horizontal grid as the atmosphere model (described in the main text). The E3SM river runoff model uses a horizontal grid mesh of ~12 km, though output from this component is not addressed in this study. The atmosphere model is initialized globally using ECMWF Reanalysis version 5 [ERA5; Hersbach et al. (2020)], while initial conditions for the land and river models are taken from the end of a 1 yr E3SM simulation forced by observed, time-evolving atmospheric data.

Our E3SM ensemble setup is guided by the approach used in Reed et al. (2020). For all E3SM simulations described in this study, the Betacast software toolkit (Zarzycki and Jablonowski, 2015, see Code Availability section for software access) is used to facilitate ensemble configuration and management. A "test" collection of 100 simulations is conducted using 10-member ensembles each initialized every 12 hours from August 27 00Z (approximately 12 hours before Hurricane Irene's first U.S. landfall in North Carolina) back to August 22 12Z. The "test" ensemble members use randomly drawn values (within defined bounds) of parameters to which hurricanes are most sensitive (see Table A1), according to the hurricane parameter sensitivity study of He and Posselt (2015) and as used in Reed et al. (2020) and Reed et al. (2021). The same parameter sets are used across each initialization time as in Reed et al. (2020). All E3SM-simulated tracks and storm characteristics (e.g., minimum sea-level pressure, maximum surface winds) of Irene are computed using the TempestExtremes software package (Ullrich et al., 2021), and model errors in storm location and intensity compared to the National Centers for Environmental Information (NCEI) International Best Track Archive for Climate Stewardship (IBTrACS; Knapp et al., 2010, 2018) are obtained using version 10.0.0 of the Model Evaluation Tools Tropical Cyclone (MET-TC) diagnostic package (Brown et al., 2021) (see Code Availability section for access to TempestExtremes and MET-TC). For each initialization time, "test" ensemble mean hurricane track and intensity errors are computed, and an optimal initialization time (in this case, August 26 00Z) is identified that attempts to maximize both simulation fidelity and forecast lead time. To create a larger sample size, the "test" ensemble initialized at the optimal time (August 26 00Z), plus the ensemble initialized 12 hours earlier (August 25 12Z), are expanded to 50 members each (Reed et al. (2020) produced a 100-member ensemble at a single initialization time). In total, two 50-member ensembles are generated by initializing E3SM at two times separated by 12 hours [00Z August 26 (hereafter "E1") and 12Z August 25 (hereafter "E2")]. Each ensemble member within E1 uses a unique parameter set, with the same parameter sets being imposed on E2 members (i.e., E2 member N uses the same parameter values as E1 member N, despite different initialization times).

Table A1: List of E3SM atmospheric physics parameters modified in this study, including the associated parameterization scheme; default values and minimum and maximum ranges used, and a short description. Parameter value ranges are taken from He and Posselt (2015). For a description of parameterization schemes, see Golaz et al. (2022).

| Scheme | Parameter | Default | Min | Max | Short Description |
|--------|-----------|---------|-----|-----|-------------------|
| CLUBB | $clubb\_c\_k10$ | 0.35 | 0.2 | 0.6 | Coefficient of momentum diffusivity, Kh_zm |

| ZM | $zmconv\_c0\_ocn$ | 0.002 | 0.001 | 0.0045 | Autoconversion coefficient over ocean for deep convection |
|---|---|---|---|---|---|
| | $zmconv\_dmpdz$ | $-0.7e^{-3}$ | -0.002 | 0. | Parcel fractional mass entrainment rate |
| | $zmconv\_tau$ | 3600. | 1800. | 28800. | Time scale for consumption rate of CAPE for deep convection |
| MG2 | $ice\_sed\_ai$ | 500 | 50 | 1400 | Cloud ice fall speed parameter |

Figure A1 displays the time evolution of errors in along- and cross-track distances, minimum central sea-level pressure, and maximum surface wind associated with Hurricane Irene simulated by E3SM initialized on August 26, 2011 00Z (ensemble E1). Similar time series for ensemble E2 (not shown) indicate larger distance errors—consistent with a more westward/inland track—but similar errors in minimum central pressure and maximum surface winds, by construction. Figure A1 shows that Hurricane Irene simulated for E1 generally follows the correct trajectory (cross-track errors less than 20 km) but has a forward speed that is slower than observed (along-track errors of roughly $-50$ to $-100$ km). Further, the E1 version of Irene predicts a central pressure that is too low and surface winds that are too high, indicating an overestimation of hurricane intensity.

We also use the Climate Prediction Center "Unified Gauge-Based Analysis of Daily Precipitation over CONUS" product (Chen et al., 2008), provided at daily resolution on a $0.25° \times 0.25°$ horizontal grid, to assess space-time averaged precipitation accumulations. Figure A2 displays the evolution of mid-Atlantic watershed-averaged (left) 3-hourly precipitation amounts and (right) cumulative sum of precipitation. Watershed-averaged precipitation intensity peaks near August 28, 2011 15Z-18Z, for both the ensemble mean and selected E1 member. E3SM exhibits a slow onset of the watershed-averaged cumulative sum of precipitation through August 28, 2011 12Z (Fig. A2, right), but later overestimates cumulative precipitation through August 29, 2011 12Z. This equates to a $\sim$15 mm ($\sim$20%) underestimation of cumulative precipitation during the initial impact window but a $\sim$30 mm ($\sim$33%) overestimation of storm-total precipitation. Together, Figs. A1 and A2 indicate that E3SM simulates a version of Irene that is too slow and too strong, leading to a delayed onset of precipitation in the mid-Atlantic watershed but ultimately an overestimation of storm-total precipitation.

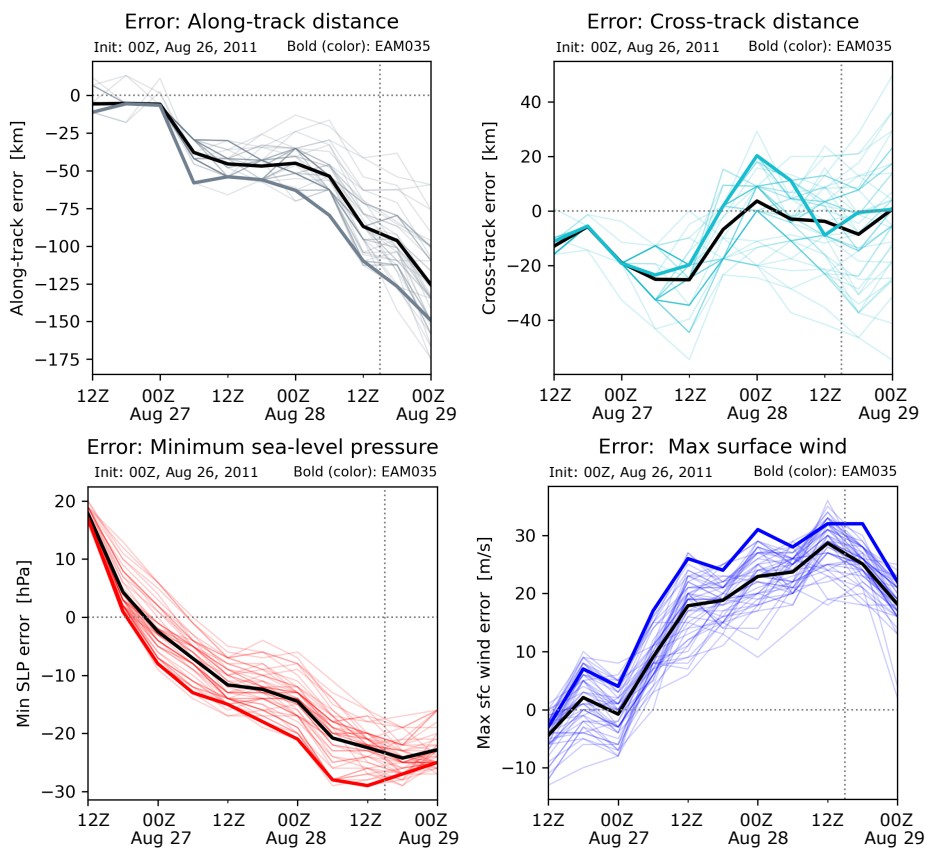

**Figure A1.** (Upper left) Time-evolving, along-track distance error (simulation minus observations) for E3SM ensemble members (light gray), the ensemble mean (bold black), and the selected ensemble member analyzed in this study (bold gray) for the Hurricane Irene initialized on August 26, 2011 00Z. (Upper right, lower left, lower right) As in the upper left panel, but for the cross-track error, minimum sea-level pressure error, and maximum surface wind, respectively. The gray dotted line marks on August 28, 2011 15Z, the time corresponding to the snapshot of Hurricane Irene shown in Fig. 2.

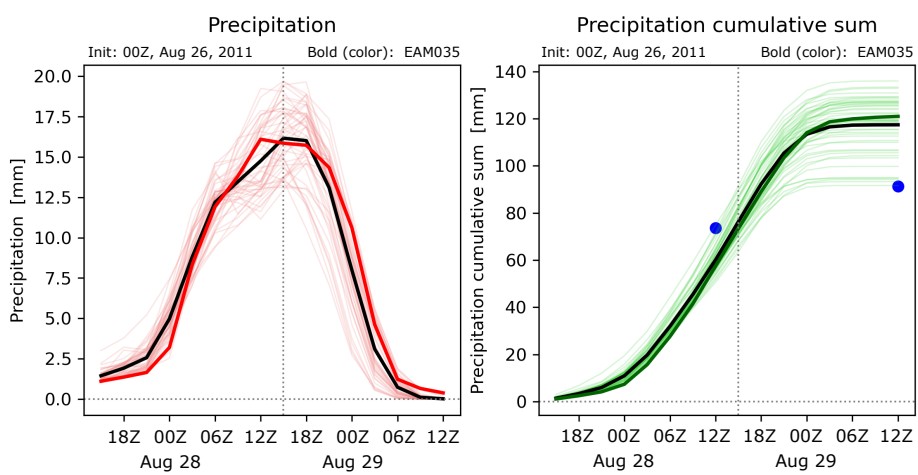

**Figure A2.** (Left) Time evolution of 3-hourly precipitation amount averaged across the mid-Atlantic HUC2 watershed for E3SM ensemble members (light red), the ensemble mean (black), and the selected ensemble member analyzed in this study (bold red) for the Hurricane Irene initialized on August 26, 2011 00Z. (Right) As in the left panel, but for the cumulative sum of watershed-averaged precipitation from August 27, 2011 12Z, through August 29, 2011 12Z, representing the time window of Irene precipitation impacts on the watershed. In the right panel, CPC rain gauge values are shown as blue dots and indicate cumulative sums for the 24-hour period ending at the time plotted.

## Appendix B: DHSVM validation

To evaluate the hydrological model for its robustness in capturing the spatial and temporal variability of flow responses, we chose Hurricane Irene (2011) as the focal event and used a gridded ($\sim 6\,km$) CONUS-scale meteorological dataset developed by Livneh et al. (2013) for model forcing. The source Livneh dataset consists of daily records of precipitation, maximum and minimum air temperature, and wind speed over the period 1950–2013, and they were disaggregated from the daily records to the 3-hourly interval. Then, we evaluated the DHSVM simulated daily flows (averaged from the 3-hourly flows) against the United States Geological Survey (USGS) daily flow observations at six gage locations. These locations represented a range of drainage areas from 4118 - 17560 sq. km along the longitudinal profile (upstream–downstream) of the main stem of the Delaware River (Figure B1a). The model performance was measured using Nash-Sutcliffe efficiency (NSE) and the Kling–Gupta efficiency [KGE; Gupta et al. (2009)]; they are commonly used for measuring hydrological model performance and can be estimated as

$$NSE = 1 - \frac{\sum_{n=1}^{N} (O_n - M_n)^2}{\sum_{n=1}^{N} (O_n - \bar{O})^2} \tag{B1}$$

Where $M_n$ and $O_n$ are simulated and observed daily flow, respectively, N is the total number of days used in metric calculations, and $\overline{O}$ is observed daily mean flow over N days.

$$KGE = 1 - \sqrt{(r-1)^2 + (\alpha-1)^2 + (\beta-1)^2} \tag{B2}$$

Where $r$ is the linear correlation, $\alpha$ is the viability error, and $\beta$ is the bias between observed and simulated daily flows. Both NSE and KGE range from $-\infty$ to 1, and a value of 1 indicates perfect agreement between simulation and observations.

The comparison between simulations and observations of daily streamflow is shown in Figure B1b. We observed a good agreement for Hurricane Irene (2011), where NSE daily range is from 0.71 - 0.91, and KGE range is from 0.59 - 0.91. Even though the model captured the timing of peak river discharge for all evaluated gages, a higher peak flow bias is observed at USGS-01428500 and USGS-01463500. Among various factors that might have attributed to this bias, some key sources are uncertainties in climate input and topography and the uncertainty in stage-discharge rating curves and streamflow records.

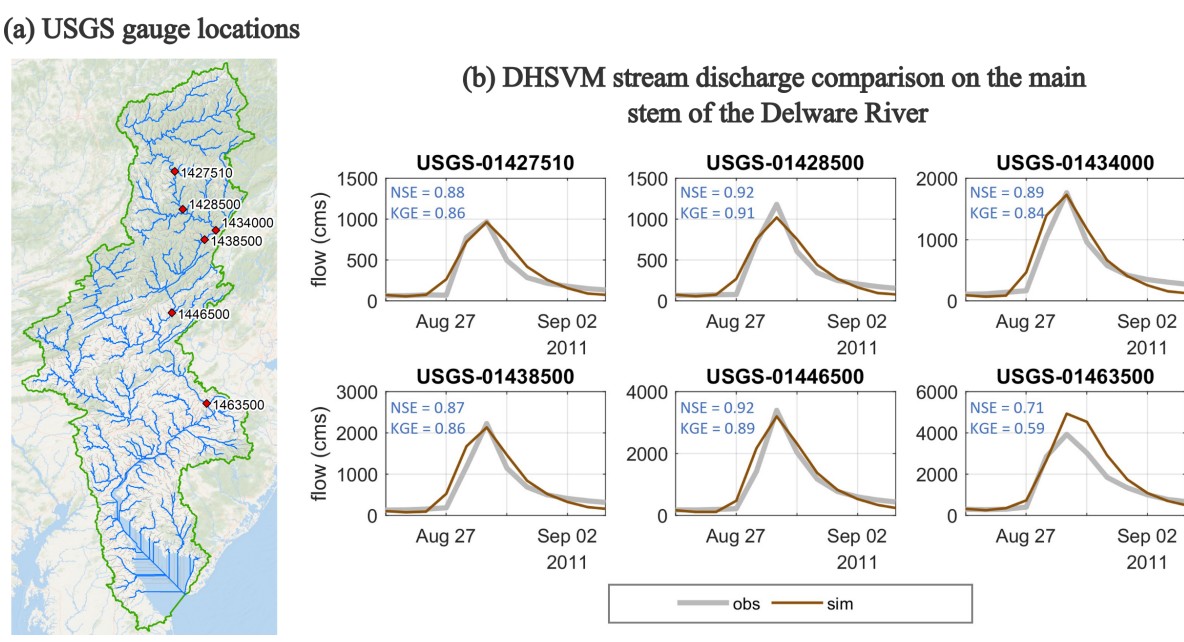

**Figure B1.** (a) USGS stream gage locations in the Delaware River Basin; (b) Comparison between observed and simulated river discharge at the USGS stream gage locations.

## Appendix C: FVCOM validation

To validate the FVCOM model results, we used a different set of model forcing than the one mentioned in the main text (section 2). The sea surface wind stress and atmospheric pressure field of Hurricane Irene (2011) were collected from ERA5, a global atmospheric reanalysis model that provided the highest resolution from the available public data – a spatial resolution of 30 km. As the hydrology model (DHSVM) was only calibrated for Hurricane Irene (2011), for validating FVCOM for both Hurricanes Irene (2011) and Sandy (2012), we assigned the river flow condition using USGS stream gages available at the Delaware River and Schuylkill River model boundary. Also, we collected water surface elevation (WSE) data from the NOAA Tides & Currents database (tidesandcurrents.noaa.gov) for model validation. Finally, to quantify FVCOM model WSE error statistics, we estimated the linear correlation coefficient (corr), average bias index (bias), and model skill as

$$corr = \frac{\sum_{n=1}^{N} \left( M_n - \bar{M} \right) \left( O_n - \bar{O} \right)}{\sqrt{\left( \sum_{n=1}^{N} \left( O_n - \bar{O} \right)^2 \right) \left( \sum_{n=1}^{N} \left( M_n - \bar{M} \right)^2 \right)}} \tag{C1}$$

$$bias = \frac{\sum_{n=1}^{N} \left( M_n - O_n \right)}{\sum_{n=1}^{N} O_n} \tag{C2}$$

$$skill = 1 - \frac{\sum_{n=1}^{N} \left( M_n - O_n \right)^2}{\sum_{n=1}^{N} \left( \left| M_n - \bar{O} \right| + \left| O_n - \bar{O} \right| \right)^2} \tag{C3}$$

Where $M_n$ and $O_n$ are simulated and observed water surface elevation, respectively, N is the total number of samples used in metric calculations, and $\overline{M}$ and $\overline{O}$ are the mean of the samples.

Figure C1 shows that the model WSE and phase match very well with the in situ WSE and show a strong ability to predict the peak surge elevation for both hurricanes. While the model peak surface amplitude agrees well, even in Philadelphia, there is a slightly elevated amplitude error after the landfall period of Hurricane Sandy (2012). This could be from the missing sub-tidal elevation at the model north-western open boundary, where we have a smaller area coverage for the ocean. The error estimates are shown in the scatter plot (Figure C2), where we only included gauges representing different estuary and river-dominant zones. Overall, the correlation coefficient varies from 0.92 to 0.96, and the skill score from 0.92 to 0.98, respectively, and the averaged bias index indicates a model underprediction due to a slightly elevated tidal damping during the flood.

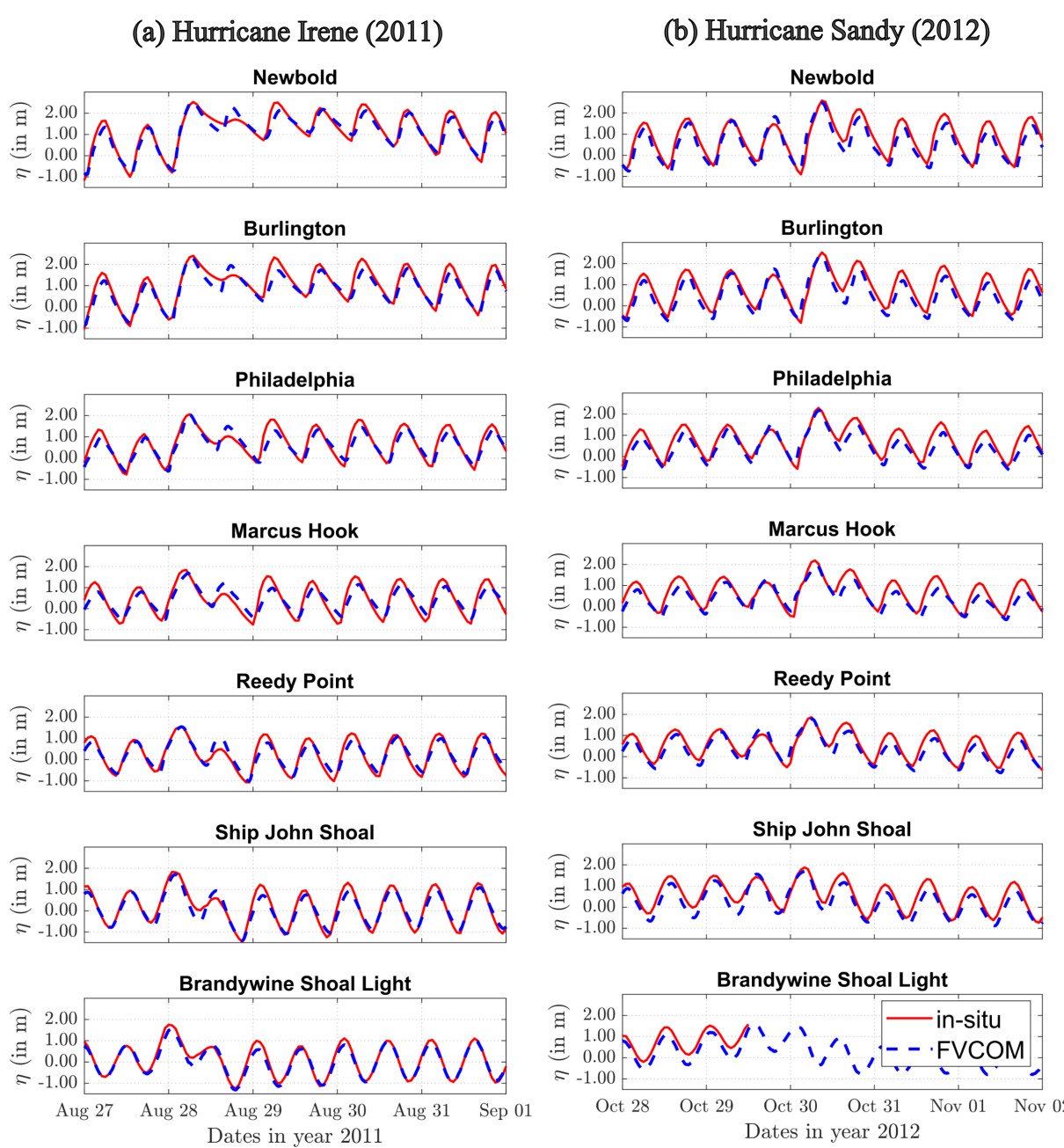

**Figure C1.** Water surface elevation comparison between model results and in situ (in meters) at NOAA tide gauge locations in Delaware Bay and River during (a) Hurricane Irene (2011) and (b) Hurricane Sandy (2012).

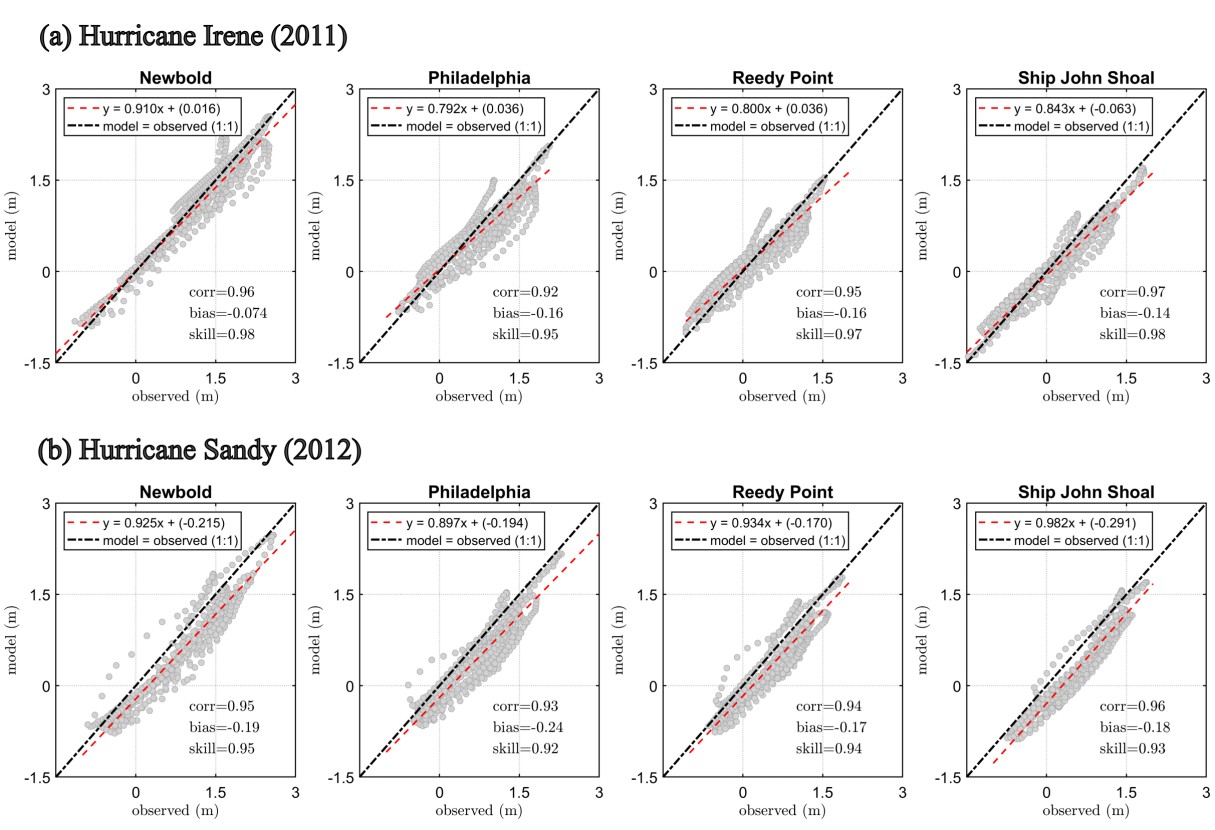

**Figure C2.** Scatter comparison and statistics for (a) Hurricane Irene (2011) and (b) Hurricane Sandy (2012) water surface elevation at four NOAA tide gauge locations that represent different bay and river zones, going from the upstream river to the estuary.

*Author contributions.* MD, ZY, and RH contributed to the concept of the study. Model simulations and data processing were conducted by MD, JB, and NS. FVCOM grid generation was worked on by MD and TW. The first version of the manuscript was drafted by MD and JB. DJ supervised the research and funding acquisition. All authors discussed the various model performances and results and revised the manuscript.

*Competing interests.* The authors report no conflicts of interest.

*Acknowledgements.* This work was supported by the MultiSector Dynamics (MSD), Office of Science, Office of Biological and Environmental Research as part of the multi-program, collaborative Integrated Coastal Modeling (ICoM) project. Also, we performed this study in collaboration with the Cross-Cutting (CC) research area of the ICoM project. All model simulations were performed using resources available through Research Computing at Pacific Northwest National Laboratory and the National Energy Research Scientific Computing Center (NERSC), a U.S. Department of Energy Office of Science User Facility located at Lawrence Berkeley National Laboratory operated under Contract No. DE-AC02-05CH11231 using NERSC award BER-ERCAP m3780 for 2022. CPC Global Unified Gauge-Based Analysis of Daily Precipitation data provided by the NOAA PSL, Boulder, Colorado, USA, from their website at https://psl.noaa.gov.

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
