# Peer review of "Estuarine hurricane wind can intensify surge-dominated extreme water level in shallow and converging coastal systems"

_EGUsphere, 2023_

## Author Comment (AC1)

We are grateful to the reviewer for the constructive comments that helped improve the manuscript. In the following, we state the referee's comments (in blue) followed by the response and actions taken (in black). We have also highlighted the changes in the revised manuscript, where new texts are represented using blue and deleted texts using red. The line numbers given here are also from the edited version.

**1 Referee 1**

**Comment #1**

Lines 038–053: While some earlier research has been provided in the INTRODUCTION section, the authors are encouraged to review and include two additional relevant publications that particularly addressed related research themes and Atlantic Ocean tropical cyclones. First, Parker et al. (2023;https://doi.org/10.1007/s11069-023-05939-6) used the GTSM-ERA5 model to investigate the proportional impacts of storm surge, wave setup, and astronomic tides on extreme water levels along the U.S. Southeast Coast. Their study revealed distinctive regional trends in the average contributions of storm surge and waves to extreme water levels during a 38-year period. Extreme water levels in the region result from combined surge, tide, and wave effects. The significance of each component varies across different locations. Next, Hsu et al. (2023; https://doi.org/10.5194/nhess-23-3895-2023) used the COAWST model to analyze how the storm characteristics of three historical hurricanes affected the storm surges and wave runup along the South Atlantic Bight. It was revealed that as lower storm translation speed prolongs total water level exceedance, potentially causing greater economic losses. Wave setup and swash predominantly impacted peak total water level, with pre-storm wave runup sometimes surpassing peak storm total water level under specific conditions.

**Response to comment #1**

We want to thank the reviewer for providing these useful references. We have included them in our introduction to clarify more about the sensitivity of extreme water levels to hurricane characteristics and the interaction of storm surge, wave setup, and astronomic tides. We also added Suh and Lee (2018) here based on the suggestion in comment #9.

On page 2, lines 41-48: *More specifically, an inaccurate representation of any of these hurricane characteristics: intensity, size, translation speed, and the angle of landfall with the coast can introduce large biases in predicting the surge and coastal flooding (Suh and Lee, 2018). In a recent study, Hsu et al. (2023) examined the role of these variables for three different hurricanes that propagated through the South Atlantic Bight [Matthew (2016), Dorian (2019), and Isaias (2020)] and showed how they affect the peak storm surge and wave runup in the South Atlantic coastline. For the same region, Parker et al. (2023) demonstrated that various combinations of tide, non-tidal residual, and wave setup and their spatially varying interaction can control the total water level at different U.S. Southeast Atlantic coastline regions.*

**Comment #2**

Lines 102–104: It would be great if the authors could explain how the total number of parameter sets (i.e., 50) was determined. Is it, for instance, based on any earlier research? Or did the authors carry out a sensitivity study?

**Response to comment #2**

We've added text to Appendix A to clarify our ensemble setup, which we'll paraphrase here: Following the methods of Reed et al. (2020; https://doi.org/10.1126/sciadv.aaw9253), we test the sensitivity of hurricane fidelity to model initialization time. This is accomplished by conducting "mini" 10-member ensembles, each initialized at 12-hour increments from just before Irene's first U.S. landfall in North Carolina back to 5 days previous. We analyze the (mini) ensemble mean hurricane track and intensity errors at each initialization time to identify an optimal initialization time that attempts to maximize both simulation fidelity and forecast lead time (to allow sufficient hurricane spin-up). As 10-member ensembles yield an inadequately small sample size, we expand the mini ensemble initialized at the optimal time (26 August 2011 00Z), as well as the ensemble initialized 12 hours earlier (25 August 2011 12Z), to 50 members each. This is very similar to the approach used in Reed et al. (2020), in which a 100-member ensemble was run at a single optimal initialization time. We felt that using a second initialization time might provide a greater diversity of model solutions – though admittedly the large inland track bias in the earlier initialization (E2) was unexpected.

**Comment #3**

Figure 2(d): The authors are encouraged to indicate the direction reference. In other words, does a vector pointing toward the top in the lower part of the figure indicate southward wind? Also, is the length of the vectors proportional to the wind speed?

**Response to comment #3**

We have added the directional reference to Figure 2(d). In this plot, the red and blue wind vectors represent the southerly and northerly wind, respectively, and they are scaled using the wind speed. We also edited the Figure 2(d) caption to explain it better.

**Comment #4**

Lines 104–107: Does this mean that the initial locations of the track/storm eye of ensemble E1 is the re-analyzed storm eye location on August 26, 2011 00:00, and the initial locations of the track/storm eye of ensemble E2 is the re-analyzed storm eye location on August 25, 2011 12:00? If yes, the authors might want to indicate it and provide more information here.

**Response to comment #4**

The reviewer is correct – the storm centers are slightly different in E1 vs. E2 because the ERA5-based meteorological patterns are slightly different between the two initialization times. We have added the following sentence to clarify this:

On page 6, lines 112-114: "Both the ERA5-based meteorological patterns and the diagnosed storm center at these two initialization times are slightly different from each other and ultimately result in a more diverse spread of simulated Irene tracks."

**Comment #5**

Lines 157–159: The authors are encouraged the make the format of date/time consistent throughout the manuscript. In lines 105, the authors used 26 August 2011 00Z and 25 August 2011 12:00Z, which were already different from each other. These date/time formats in lines

105 are also different from the format in lines 157–159. The authors are also encouraged to make the date/time formats on figure axes consistent throughout the manuscript (e.g., Figures A1, A2, and B1).

**Response to comment #5**

We have modified all the date/time formats within the main text and on figure axes to make them consistent.

**Comment #6**

Lines 164–166: The authors are encouraged to provide more details about why Cases A and B were selected specifically from ensembles E1 and E2. For instance, do they have the highest peak surges or largest wind speed at specific locations compared to other cases in the two ensembles? In addition, the authors are encouraged to quantitatively demonstrate how similar are the surges produced by Cases A and B. For example, does this statement imply that the difference in peak surge elevations is less than a particular amount during a specific period? Or does this imply that the root-mean-square difference (or any other indicators) between the two time series is smaller than any given number?

**Response to comment #6**

We previously did not discuss the selection process of Cases A and B in detail in the main text. To address this concern, we now include a supplemental material section "Hurricane case selection: Case A from Ensemble E1 and Case B from E2" that shows the step-by-step approach we took before selecting these two tracks. We added the following paragraph in that supplemental material section.

*One of the main objectives of this study is to demonstrate how estuarine wind fields can exacerbate hurricane-driven coastal and riverine flooding. To illustrate this, we chose two cases from the different ensembles of hurricanes E1 and E2 that can provide a more straightforward explanation of the variation in along-channel peak water surface elevation (WSE) from the interaction of surface wind stress inside the estuary, tide, and geometry, shown in Figure 2c. In Figure 2c, between SJS and RP (close to 80 km from the bay entrance), we can notice a higher peak WSE for E1. This location is near the open bay, far from the influence of river discharge, and the remote- and estuarine-wind-generated surges primarily dominate the flooding. Because of the largest bay WSE there (in Figure 2c), we compared the time series of WSE between all members of E1 (shown in Figure S1) and observed that member/case 35 produced the highest WSE. Hence, we picked this event from E1 and called "Case A" (which has a primarily northerly wind) to further assess the role of the estuarine local wind. Compared to E1, E2 has more inland tracks (primarily southerly wind) and generates a much higher range of surges, especially in the mid-bay. To explain the role of the estuarine local wind in E2 surges, we tried to rank the members that have negligible influence from river discharge on WSE, have a similar magnitude of wind speed, time scale (translation through the bay), and WSE at the bay entrance compared to Case A from E1. Identifying such an event from E2 can easily demonstrate the role of the southerly estuarine wind in surge amplification. Figure S2 compares Delaware River discharge at the model flow boundary where we identified four members, 42, 22, 8, and 14, that produced the lowest fluvial discharge. Subsequently, we also examined the wind speed and direction from these E2 members with Case A from E1 (shown in Figure S3; the red and blue wind vectors represent the southerly and northerly wind, respectively, and they are scaled*

*using the wind speed) and observed that E2 Case 14 generated a relatively similar wind speed and time scale than others. Before choosing Case 14 from E2 for further analysis, we also compared the WSE at the bay entrance. While we did not compare the root-mean-square of the four members from E2 with Case A because of the phase lag between E1 and E2 members, from Figure S4, we can see that Case 14 has the closest peak WSE to Case A among them and the difference is around 0.27 m. Ultimately, based on all these different comparisons, we picked E2 Case 14 as "Case B" to demonstrate the role of estuarine southerly wind direction in elevating the surge-induced flooding in Delaware Bay and River compared to northerly wind direction or "Case A".*

On page 10, line 176 of the main article:

*We provided a more thorough description of different process comparisons that led to these event selections in the supplemental material titled "Hurricane case selection".*

**Comment #7**

Lines 170–172: Does this mean that, regardless of the storm eye's instantaneous location, all computational grids within the polygon must have zero wind speeds? Furthermore, it is recommended that the authors explain the process by which this polygon was created and established. For instance, will the results change significantly if the shape of this polygon changes?

**Response to comment #7**

Section 4 now provides more details on this force modification.

On page 10, line 184: *We used this bounding polygon to select the E3SM grid cells that fully cover the FVCOM model domain. FVCOM uses a bilinear interpolation method to assign wind velocity at the unstructured grid cells from the meteorological dataset. To represent a scenario with nominal estuarine wind during hurricane landfall, we multiplied the E3SM wind velocity vectors (in m/s) within the polygon with 0.1 to uniformly dampen the wind magnitude in the selected cells, regardless of the instantaneous location of the hurricane. For the two cases, A and B, this artificial dampening reduced the peak wind speed magnitude to $\sim 2.0\,m/s$, making the impact of the hurricane wind field negligible. In addition, the polygon and E3SM grid cells extensively covered the FVCOM model domain, going beyond the FVCOM boundary, to better interpolate the wind forcing.*

On page 11, line 214: *As described earlier, we multiplied the E3SM wind velocity vectors (in m/s) by 0.1 outside/within the polygon (shown in Figures 3a,b) to make estuarine/remote wind stress-only cases, respectively.*

Furthermore, we would like to state that the results will not change based on the shape as long as it contains all the E3SM cells that overlap the FVCOM unstructured grid cells representing the bay and the river (wet cells).

**Comment #8**

Lines 197–199: While it is understood that the gradients of the curves look 'similar' by eye, the authors are encouraged to quantitatively describe how similar are the along-channel peak WSE gradients. For example, root-mean-square difference or model skill might be good indicators.

**Response to comment #8**

We added the following lines to the main text to clarify the difference between the along-channel peak WSE for the two cases.

On page 11, line 219: *When the full wind field is included, we can see a similar along-channel peak WSE gradient where the difference in peak water level, varying from 0.73 to 1.05 m, came from the remote surge propagation through Delaware Bay.*

**Comment #9**

Lines 223–260: The authors are encouraged to include some more relevant studies in the discussion section. In addition to the works of Parker et al. (2023) and Hsu et al. (2023) mentioned above, Suh and Lee (2018; https://doi.org/10.1016/j.csr.2018.09.007) analyzed the effects of storm translation speed and storm path on surge propagation processes and surge level along the coast. The authors might also want to consider including Suh and Lee (2018) in the INTRODUCTION. Incorporating discussions on these relevant works in the DISCUSSION AND CONCLUSIONS section is thought to be advantageous for readers, offering a more comprehensive background.

**Response to comment #9**

We have included the necessary references in the discussion section to briefly explain what was lacking in the previous studies, how we addressed the interaction between estuarine wind and surge using a physics-based integrated framework, and the future tasks that need to be done.

On page 15, line 250: *Previous works related to the sensitivity of storm surge and coastal flooding to hurricane landfall locations, wind field (speed and direction), and geometry [e.g., Powell and Houston (1996); Houston et al. (1999); Shen et al. (2006b); Weisberg and Zheng (2008); Marsooli and Lin (2018)] have not separately examined the role of this shorter period (translation period through the estuary) and estuarine-scale landfalling wind using physics-based integrated modeling frameworks.*

On page 15, line 256: *Further work is also needed to examine the role of hurricane intensity, the radius of maximum wind, translation speed, and the interaction between tide, non-tidal residual, and waves, separately, all of which could similarly influence the coastal flood level (Suh and Lee, 2018; Parker et al., 2023; Hsu et al., 2023).*

---

## Author Comment (AC2)

We want to thank the reviewer for the constructive comments. In the following, we state the referee's comments (in blue) followed by the response and actions taken (in black). We have also highlighted the changes in the revised manuscript, where new texts are represented using blue. The line numbers given here are also from the edited version.

**1 Referee 2**

**Comment #1**

The selection of Hurricane Irene (2011) is very appropriate due to the severity and the influence on the study site. However, did the authors consider the option of using other hurricane tracks instead of perturbing Hurricane Irene or to complement the results obtained with Hurricane Irene?

**Response to comment #1**

We selected the Delaware Bay and River (DBR), US, as the study site to examine the role of local estuarine wind in elevating the surge-induced coastal flooding because of the extensive model calibration and validation done for both hydrology (DHSVM) and hydrodynamic (FVCOM) models in the same region previously (using a large set of available flow gauges, tide gauges, tidal current profilers, and high water marks), described in Deb et al. (2023; https://doi.org/10.1029/2022EF002947). For this region, it has been reported that Hurricane Irene (2011) caused the most damage when both fluvial and coastal flooding are considered. Two other recent extreme events, Hurricane Isabel (2003) and Hurricane Sandy (2012), also brought large coastal surges within the bay and river; however, they made landfall far away from DBR and translated through different regions, making them less suitable cases compared to Hurricane Irene (2011).

**Comment #2**

Regarding the generation of the ensembles, please justify the selection of 50-member ensembles and the two times selected for the initialization (separated by 12 hours). Please, discuss the sensitivity of the results/conclusions to these parameters.

**Response to comment #2**

Reviewer #1 posed a very similar question. We've added text to Appendix A to clarify our ensemble setup, which we'll paraphrase here: Following the methods of Reed et al. (2020; https://doi.org/10.1126/sciadv.aaw9253), we test the sensitivity of hurricane fidelity to (a) model physics parameters and (b) model initialization time. With regard to (a), a suite of model simulations (to be explained below) are conducted by perturbing model physics parameters to which hurricanes are most sensitive, according to the hurricane parameter sensitivity study of He and Posselt (2015; https://doi.org/10.1175/JCLI-D-15-0255.1) and as used in Reed et al. (2020) and Reed et al. (2021; http://dx.doi.org/10.1175/BAMS-D-20-0160.1). With regard to (b), this is accomplished by conducting "mini" 10-member ensembles, each initialized at 12-hour increments from just before Irene's first U.S. landfall in North Carolina back to 5 days previous. We analyze the (mini) ensemble mean hurricane track and intensity errors at each initialization time to identify an optimal initialization time that attempts to maximize both simulation fidelity and forecast lead time (to allow sufficient hurricane spin-up). As 10-member ensembles yield an inadequately small sample size, we expand the mini ensemble initialized at

the optimal time (00Z August 26, 2011), as well as the ensemble initialized 12 hours earlier (12Z August 25, 2011), to 50 members each. This is very similar to the approach used in Reed et al. (2020), in which a 100-member ensemble was run at a single optimal initialization time. We felt that using a second initialization time might provide a greater diversity of model solutions – though admittedly the large inland track bias in the earlier initialization (E2) was unexpected.

**Comment #3**

The validation of the FVCOM model (Appendix C) shows good results. However, the validation was carried out by forcing the model with ERA5 instead of E3SM outputs. Why was the FVCOM model not validated with E3SM outputs? It is suggested to include the validation of the hydrodynamic model with the forcing data used in the analysis or discuss in more detail this issue in the paper. This comparison would be also useful for understanding the uncertainty that justifies the generation of the 50-member ensembles.

**Response to comment #3**

This work first validated hydrology and hydrodynamic models using observation-based and reanalysis forcing to show that they can accurately represent the physical processes if we provide a more realistic forcing. Then, we used the E3SM ensembles of Hurricane Irene-like events to examine the sensitivity of coastal flooding to different tracks. In response to the reviewer's question "*Why was the FVCOM model not validated with E3SM outputs?*", we want to say that the E3SM tracks used in this study represent Irene-like events which are all physically plausible; however, they do not precisely represent the observed event. In Appendix A, we have an in-depth discussion about the time evolution of errors in along- and cross-track distances, minimum central sea-level pressure, and maximum surface wind associated with the E3SM tracks compared to the observed event. See the following description:

On page 17, line 318: *Figure A1 displays the time evolution of errors in along- and cross-track distances, minimum central sea-level pressure, and maximum surface wind associated with Hurricane Irene simulated by E3SM initialized on August 26, 2011 00Z (ensemble E1). Similar time series for ensemble E2 (not shown) indicate larger distance errors—consistent with a more westward/inland track—but similar errors in minimum central pressure and maximum surface winds, by construction. Figure A1 shows that Hurricane Irene simulated for E1 generally follows the correct trajectory (cross-track errors less than 20 km) but has a forward speed that is slower than observed (along-track errors of roughly −50 to −100 km). Further, the E1 version of Irene predicts a central pressure that is too low and surface winds that are too high, indicating an overestimation of hurricane intensity. We also use the Climate Prediction Center "Unified Gauge-Based Analysis of Daily Precipitation over CONUS" product (Chen, 2008), provided at daily resolution on a 0.25°×0.25° horizontal grid, to assess space-time averaged precipitation accumulations. Figure A2 displays the evolution of mid-Atlantic watershed-averaged (left) 3-hourly precipitation amounts and (right) cumulative sum of precipitation. Watershed-averaged precipitation intensity peaks near August 28, 2011 15Z-18Z, for both the ensemble mean and selected E1 member. E3SM exhibits a slow onset of the watershed-averaged cumulative sum of precipitation through August 28, 2011 12Z (Fig. A2, right), but later overestimates cumulative precipitation through August 29, 2011 12Z. This equates to a ∼15 mm (∼20%) underestimation of cumulative precipitation during the initial impact window but a ∼30 mm (∼33%) overestimation of storm-total precipitation. Together, Figs. A1 and A2 indicate that E3SM simulates a version of Irene that is too slow and too strong, leading to a delayed onset of precipitation in the mid-Atlantic watershed but ultimately an overestimation of storm-total precipitation.*

To show that E1 members (with correct trajectory) can provide a reasonable estimate of the peak water surface elevation (WSE) when compared to observation, we have modified Figure 2c and included the observed peak WSE during Hurricane Irene (2011). This modified Figure 2c can help understand the uncertainty in FVCOM water elevation due to varying biases in meteorological forcing.

We also added the following lines to the main text:

On page 7, line 157: *E1, which produced reasonable Irene tracks (Figure A1), also shows a fair range of WSE and an ensemble mean compared to the observed along-channel peak WSE for Hurricane Irene (2011). Here, observed peak WSE means the FVCOM model WSE generated using reanalysis forcing and validated using field datasets. At the upstream end, near NB, the peak WSE range deviates from the observed due to the influence of river discharge and the biases that propagated from the E3SM precipitation field (details provided in Appendix A).*

**Comment #4**

In Fig.3 and several parts of the document, the authors mention 'simulations without the effect of estuarine wind fields, remote wind fill and full wind fill'. It is not clear how these types of wind fields (estuarine, remote and will) are considered in the numerical model. For example, does a simulation without estuarine wind fields refer to a simulation with astronomical tides only or with astronomical tides and pressure (but without wind)? For a simulation with remote wind fields, how are the estuarine wind fields switched off?. Please, explain in more detail.

**Response to comment #4**

Reviewer #1 also asked a very similar question. We multiplied the E3SM wind velocity vectors (in m/s) by 0.1 outside/within the polygon (shown in Figures 3a,b) to make estuarine/remote wind stress-only cases, respectively. The astronomical tide and pressure field were not modified. We are paraphrasing the new lines below that are added to Section 4.

On page 10, line 184: *We used this bounding polygon to select the E3SM grid cells that fully cover the FVCOM model domain. FVCOM uses a bilinear interpolation method to assign wind velocity at the unstructured grid cells from the meteorological dataset. To represent a scenario with nominal estuarine wind during hurricane landfall, we multiplied the E3SM wind velocity vectors (in m/s) within the polygon with 0.1 to uniformly dampen the wind magnitude in the selected cells, regardless of the instantaneous location of the hurricane. For the two cases, A and B, this artificial dampening reduced the peak wind speed magnitude to $\sim 2.0\,m/s$, making the impact of the hurricane wind field negligible. In addition, the polygon and E3SM grid cells extensively covered the FVCOM model domain, going beyond the FVCOM boundary, to better interpolate the wind forcing.*

On page 11, line 215: *As described earlier, we multiplied the E3SM wind velocity vectors (in m/s) by 0.1 outside/within the polygon (shown in Figures 3a,b) to make estuarine/remote wind stress-only cases, respectively.*

**Comment #5**

The results are mainly based on the comparison of cases A and B. How are the remaining Ensemble 1 and 2 simulations used in the analysis?

**Response to comment #5**

In this study, we used all the Ensemble 1 and 2 simulations at the beginning to see the variability in water surface elevation from different events (shown in Figure 2c). Subsequently, as we focused on examining the primary driving mechanism behind the increase in along-channel water surface elevation, especially at the mid-Bay, we selected these two cases, A and B, which represented the best combination from all members to show the role of the estuarine local wind in flood amplification. We added a new supplemental material section "Hurricane case selection: Case A from Ensemble E1 and Case B from E2", based on Reviewer #1's comment, showing the step-by-step approach we took before selecting these two tracks.

**Comment #6**

The study focuses on convergent estuaries, can similar results and conclusions be expected for other types of estuaries? I would suggest the authors explain this in more detail in the discussion section.

**Response to comment #6**

The geometry can vary widely for other estuaries (e.g., sheltered tidal lagoons or river deltas), influencing the flood amplification or attenuation rate depending on the estuary's unique characteristics. From the results of this study, it would be challenging to state the response of these systems. However, based on the existing literature [e.g., Khojasteh et al. (2021); https://doi.org/10.1371/journal.pone.0257538], we can say that the along-channel flood amplification rate will always be higher for a converging and macro-tidal system like the Delaware Bay and River, USA, than for non-converging ones. We added the following lines to the main text to clarify it more.

On page 16, line 281: *In converging estuarine systems worldwide [e.g., the Delaware Bay and River (USA), Humber Estuary (UK), Hooghly Estuary (India), the Meghna River Estuary (Bangladesh), and the Pearl River Estuary (China)] that are highly vulnerable to hurricane-induced flooding, physically consistent and integrated modeling frameworks are critical to correctly resolve this nonlinear tide-wind-surge dynamics and improve the coastal hazard projections for a future climate. In other coastal systems, such as sheltered tidal lagoons or river deltas, properly resolving the estuarine local wind using an integrated framework is essential as well; however, the interacting effect of geometry and tide-wind-surge dynamics in flood amplification will be less significant than the converging ones.*